# Evaluation of Precision and Sensitivity of Back Extrusion Test for Measuring Textural Qualities of Cooked Germinated Brown Rice in Production Process

**DOI:** 10.3390/foods12163090

**Published:** 2023-08-17

**Authors:** Kannapot Kaewsorn, Pisut Maichoon, Pimpen Pornchaloempong, Warawut Krusong, Panmanas Sirisomboon, Munehiro Tanaka, Takayuki Kojima

**Affiliations:** 1Department of Agricultural Engineering, School of Engineering and Innovation, Rajamangala University of Technology Tawan-Ok, Chon Buri 20110, Thailand; kannapot_ka@rmutto.ac.th; 2Department of Agricultural Engineering, School of Engineering, King Mongkut’s Institute of Technology Ladkrabang, Bangkok 10520, Thailand; palmpisut1994@gmail.com; 3Department of Food Engineering, School of Engineering, King Mongkut’s Institute of Technology Ladkrabang, Bangkok 10520, Thailand; pimpen.po@kmitl.ac.th; 4Division of Fermentation Technology, School of Food Industry, King Mongkut’s Institute of Technology Ladkrabang, Bangkok 10520, Thailand; warawut.kr@kmitl.ac.th; 5Laboratory of Agricultural Production Engineering, Faculty of Agriculture, Saga University, 1 Honjo-machi, Saga 840-8502, Japan; kojimat1733@gmail.com

**Keywords:** germinated brown rice, cooked rice, texture, back extrusion, precision, sensitivity

## Abstract

The textural qualities of cooked rice may be understood as a dominant property and indicator of eating quality. In this study, we evaluated the precision and sensitivity of a back extrusion (BE) test for the texture of cooked germinated brown rice (GBR) in a production process. BE testing of the textural properties of cooked GBR rice showed a high precision of measurement in hardness, toughness and stickiness tests which indicated by the repeatability and reproductivity test but the sensitivity indicated by coefficient of variation of the texture properties. The findings of our study of the effects on cooked GBR texture of different soaking and incubation durations in the production of Khao Dawk Mali 105 (KDML 105) GBR, as measured by BE testing, confirmed that our original protocol for evaluation of the precision and sensitivity of this texture measurement method. The coefficients of determination (R^2^) of hardness, toughness and stickiness tests and the incubation time at after 48 hours of soaking were 0.82, 0.81 and 0.64, respectively. The repeatability and reproducibility of reliable measurements, which have a low standard deviation of the greatest difference between replicates, are considered to indicate high precision. A high coefficient of variation where relatively wide variations in the absolute value of the property can be detected indicates high sensitivity when small resolutions can be detected, and vice versa. The sensitivity of the BE tests for stickiness, toughness and hardness all ranked higher, in that order, than the sensitivity of the method for adhesiveness, which ranked lowest. The coefficients of variation of these texture parameters were 31.26, 20.59, 19.41 and 18.72, respectively. However, the correlation coefficients among the texture properties obtained by BE testing were not related to the precision or sensitivity of the test. By obtaining these results, we verified that our original protocol for the determination of the precision and sensitivity of food texture measurements which was successfully used for GBR texture measurement.

## 1. Introduction

Germinated brown rice (GBR) has better texture, nutritional and nutraceutical qualities, compared with brown rice [1,2,3,4]. During the GBR soaking process, water is absorbed rapidly and a series of biochemical processes occur, which result in the softening of the texture, degradation of polymers, and enhancement of the synthesis and accumulation of certain phytochemical compounds [1,5,6,7]. GBR production in a typical small-scale commercial setting may be summarized as follows: First, rough rice is soaked in water at room temperature for 48 h. The water is changed every 4 h and drained at the end of soaking to prevent fermentation of the rice. The rice is kept in a polypropylene sack for incubating at 24 h to obtain the germinated rough rice (GRR). The GRR is then dried using the fluidized bed technique by means of a superheated steam which reduces the moisture content of the rice to around 19%wb. Then, the rice is spread on screens for 3 h at room temperature, and the moisture content declines to about 13–14%wb. Finally, the GRR sample is dehusked to obtain GBR. In addition, we note here the results of a published study, which found that the initial soaking of RD31 rough rice resulted in an increase in moisture content from 12.39–13.42% (dry basis) to 24.95–35.63% (dry basis), although the moisture content only increased with time [8].

Due to its slight flavor, the textural properties of cooked rice—including cooked GBR—are dominant when assessing eating quality. Along with aroma, texture is the most important attribute for the eating quality of cooked rice. Wang et al. [9] reported a large variance of textural attributes, and a total observed number of 39 major volatile organic components, in varieties of Yueguang cooked rice. Pearson correlations showed that the hardness of cooked rice was positively correlated with the content levels of E-2-hexenal, 2-hexanol-monomer, 1-propanol, and E-2-pentenal, while stickiness was positively correlated with 5-methyl-2-furanmethanol and dimethyl trisulfide. Possible mechanisms explaining these relations were discussed in this report [9]. Research findings like these could help the rice industry to develop rice products with the desirable properties of both texture and aroma. Other studies have reported changes in the texture of cooked GBR obtained by different processes. The authors of [10] reported that brown rice with texture that was 32% and 42% softer could be produced by germination and by early harvesting, respectively [10]. Researchers have also reported that, during soaking in water, the germ of GBR produces substances both physiologically and through the action of enzymes which improve the nutritional value and the texture of brown rice [11,12]. Zhu et al. [13] investigated the effects of hydrogen-rich water (HRW) on the germination efficiency and texture of GBR and found that HRW (1.5 mg L^−1^) treatment significantly increased the germination efficiency of GBR by changing the migration of water during soaking and also improved the texture of GBR due to changes in the ultrastructure of the bran layer and a decrease in insoluble dietary fiber. In addition, subjecting germinated brown rice to microwave precooking and freeze-drying treatments made the brown rice softer, with hardness reduced by about 54.78% after treatment, and also easier to chew, as the chewiness of the precooked brown rice was lowered [14].

Cheevitsopon et al. [15] reported that an Asian Institute of Technology research group developed and used back extrusion (BE) testing [16,17] for measuring the hardness of cooked rice while following the rice cooking method described by Reyes and Jindal [16], Srisawas and Jindal [18], and Parnsakhorn and Noomhorm [19]. A BE sample holder consists of a stainless-steel cylinder of 80 mm length with an internal diameter of 15 mm and a stainless-steel spherical probe of 12.7 mm diameter. There is a 1.15 mm gap between the spherical probe with the wall of the cylinder. In the above-mentioned study, 3 g of cooked rice was placed into the cylinder without any exertion of pressure. Then, the spherical probe was moved downwards and pressed the rice sample in the cylinder at a speed of 1 mm s^−1^ until the spherical probe was 1 mm from the bottom of the cylinder when it stopped moving and returned to its initial position. BE testing followed the methods of Sirisoontaralak and Noomhorm [17], and hardness, adhesiveness and stickiness were all measured. The texture of cooked rice was measured under various measurement conditions using an extrusion test which better predicts values of sensory texture characteristics, as well as tests of Pearson correlations between the maximum force and the gradient value, or the maximum force and the area value, under each measurement condition; these showed high correlations of 0.90 or more [20]. These texture properties were illustrated in the force-time curve obtained by the test. The maximum force indicated hardness, the gradient value indicated firmness, and the area value indicated stickiness. Parnsakhorn and Langkapin [21] used the BE test for measuring the hardness of Zongzi products cooked with white glutinous rice, black sticky rice and riceberry rice as ingredients. Their first sample consisted of 100% white glutinous rice; the second sample was 50% white glutinous rice mixed with 50% black sticky rice; the third sample was 50% white glutinous rice mixed with 50% riceberry rice; and the fourth sample was 50% black glutinous rice mixed with 50% riceberry rice. 

Germinated brown rice is more popular for consumption because of its high nutritional value. The texture quality of germinated brown rice is important for consumers and, therefore, the rice industry. Rice incubation is a complicated process, which involves changes in the physical and chemical properties of the rice grain. Starch, protein and lipids are the main rice grain components which affect cooking and eating quality [22]. In addition, rice incubation commences prior to harvest and continues into the postharvest storage period. It involves dramatic changes in the physical and physicochemical properties of the rice grain, including cooking, pasting and thermal properties [23].

The United States Department of Agriculture reported in 2023 that Thailand is a globally important producer and exporter of rice. In the 2020–2021 growing season, Thailand ranked third globally with an 11.9% share of global markets, behind India (a 38.9% share) and Vietnam (12.9%), and followed by Pakistan, the USA and China [24]. Khao Dawk Mali 105 (KDML 105) is the best-known fragrant rice (*Oryza sativa*, L.) variety produced in Thailand. It is exported worldwide and is also the most-used variety for the production of GBR in Thailand. It is included in “Khao Hom Mali” in Thai. This is why we used it as the main variety in this present study.

The objective of this work was to evaluate the effect of different soaking and incubation durations in the production of Khao Dawk Mali 105 (KDML 105) GBR on the texture of cooked GBR, as measured by BE testing. We sought also to determine the precision and sensitivity of the BE test on texture properties of 32 brands of cooked GBR rice in Thailand produced by varieties of rough rice; therefore, an original protocol for the determination of precision and sensitivity of food texture measurement is proposed.

## 2. Materials and Methods

### 2.1. Samples

The GBR samples were prepared by a factory of the P.J. Brand in Chonburi Province, Thailand, and purchased from local markets in Thailand. Rough rice of the *Oryza sativa* L. cultivar Khao Dawk Mali 105 (KDML 105) was collected from a field of the P.J. Brand germinated rough rice factory in Chonburi Province, Thailand. The GBR was produced by the method used by the company, which was previously reported by Kaewsorn and Sirisomboon [25]. In brief, rough rice was soaked in water at room temperature for 24 or 48 hours to create the GBR. Every four hours, the water was changed, and it was drained at the end of the soak. To produce the germinated rough rice (GRR), the rice was incubated for seven different incubation times (0, 6, 12, 18, 24, 30 and 36 h). The fluidized-bed method was used to dry the GRR, bringing the rice’s moisture content down to about 19%wb. The moisture content of the rice was then reduced to about 13–14%wb after being spread out on a screen where the air could flow through for 3 hours at room temperature. The GRR sample was then dehusked prior to the experiment. This will be referred to as GBR in this paper. In addition, 32 commercial types and brands of germinated brown rice, some with single varieties and others with mixed varieties, as indicated in Kaewsorn and Sirisomboon [25], were purchased from local department stores in Bangkok, Thailand and stored in the laboratory at room temperature. These commercial GBR products were as follows: (m1) Khao Dawk Mali 105; (m2) Khao Dawk Mali 105; (m3) Thai jasmine rice; (m4) Thai jasmine rice; (m5) Khao Dawk Mali 105; (m6) Khao Dawk Mali 105; (m7) Thai jasmine rice; (m8) Khao Dawk Mali 105; (m9) Khao Dawk Mali 105; (m10) Thai jasmine rice; (m11) Thai jasmine rice; (m12) Khao Dawk Mali 105; (m13) Khao Dawk Mali 105; (m14) Khao Dawk Mali 105; (m15) Thai jasmine rice; (m16) Thai jasmine rice; (m17) Thai jasmine rice, Red Hawm Rice; (m18) Thai jasmine rice, Red Hawm Rice; (m19) Thai jasmine rice, Red Hawm Rice; (m20) Thai jasmine rice, Red Hawm Rice; (m21) Khao Dawk Mali 105, Red Hawm Rice, Hom Dang Sukhothai, Sinlek Rice, Homnil, Riceberry, Thai Pathumthani fragrant rice, black sticky rice, RD6; (m22) Khao Dawk Mali 105, Homnil, Red Cargo rice; (m23) species not specified; (m24) Doi rice, Red Doi rice, Kum Doi Rice, Saren Rice, Ubon Ratchathani Hommali Rice; (m25) Khao Dawk Mali 105, Red Hawm Rice, RD6; (m26) Muser Purple Rice, Red Hawm Rice, RD6, Khao Dawk Mali 105; (m27) Thai jasmine rice, Red Hawm Rice, Homnil; (m28) Red Hawm Rice; (m29) Homnil; (m30) Homnil; (m31) Red Cargo rice; (m32) Muser Purple Rice. 

### 2.2. Rice Cooking Method

The rice cooking method followed that described by Sirisomboon et al. [26]. Home electronic rice cookers (RC-10 MM, Toshiba, Thailand) were used to cook 200 g GBR samples using the water-to-rice ratio of 1.6:1 recommended for GBR by rice producers. By such means, cooked rice was obtained with a texture like that of the cooked rice typically consumed by consumers. The cooked GBR was placed into plastic cups containing approximately 5 g of product. In total, 5 cups per sample were prepared at one cooking time.

### 2.3. Back Extrusion Test

The cooked GBR samples were then subjected to the back extrusion test using 3 g of cooked rice placed into a back extrusion test rig (BE) (Figure 1) which was compressed from the top opening of the rice container by a stainless-steel ball for 99 mm of the total height of 100 mm, with a ball probe speed of 1mms^−1^. Mean values for each sample were obtained from 5 replicate measurements. The hardness, toughness, stickiness and adhesiveness of cooked GBR were determined by observing the force–time curve and recording the maximum compression force (N) (point H), the area under the curve AHB (Ns which was converted to Nmm), the negative force (N) (point C), and the area BCD (Ns which was converted to Nmm). The hardness of cooked GBR indicates the hardness or softness of the rice. This was measured when the ball probe passed through 3 g of cooked rice and reached 1 mm above the bottom of the cylinder where the cooked rice grains were crushed to the greatest degree, while the texture meter was still in safe mode. The toughness of cooked GBR is the texture parameter which indicates the ability of the cooked rice to resist the stress applied to it by the ball probe. This stress deforms the cooked GBR from the beginning of the compression until the probe reaches the bottom of the cylinder, as in case of the hardness measurement. In the present study, toughness was indicated by the extent of the area under curve from the beginning of the compression until the probe was withdrawn from the cooked rice, i.e., the amount of energy required to deform 3 g of cooked rice. The stickiness of cooked GBR was measured by detecting the maximum negative force exerted when the ball probe was being withdrawn but the deformed sticky cooked rice remained attached to the probe, thereby exerting a pulling force (the opposite to a compressing force; therefore, a negative force) upon the probe. The adhesiveness of cooked GBR was measured by calculating the negative value of the area during withdrawal of the probe when deformed sticky cooked rice remained attached to the probe and the rice exerted a pulling force upon the probe until the rice was separated from the probe. Ninety-two samples were subjected to the back extrusion test, and an average value of 5 replications was recorded for each sample.

### 2.4. The Repeatability and Reproducibility of the Measurements of Texture Properties

The repeatability and reproducibility of the measurement of texture properties were determined by measuring four duplicates (four pairs) that were randomly selected at different times during the experiment. Reproducibility was defined as the standard deviation of the differences observed between blind duplicate values. In addition, the repeatability of the reference test was determined as the standard deviation of the differences between the values obtained from four duplicates (four pairs) that were not blind samples. The repeatability indicates the precision of the analysis of the measurement methods and reproducibility indicates precision of the analyst practice.

### 2.5. Statistical Analysis

The means and standard deviations associated with the five replicates were calculated for hardness, toughness, stickiness and adhesiveness. The sensitivity of the test for each of the texture parameters was indicated by the coefficient of variation (%). One-way ANOVA was used to determine significant differences among the means for varieties of commercial GBR, and a mean comparison was obtained using Duncan’s multiple range test with a confidence level of 95%.

## 3. Results and Discussion

### 3.1. The Repeatability and Reproducibility of the Measurements of Texture Properties

Table 1, Table 2, Table 3 and Table 4 show the precision of the results of the reference testing of the textural properties of cooked GBR, i.e., hardness, toughness, stickiness and adhesiveness. It was obvious that the reproducibility was higher than repeatability because the blind samples were in the reproducibility test. The repeatability and reproducibility of the hardness, toughness and stickiness tests did not differ greatly compared with the corresponding results for the adhesiveness tests, indicating the lower precision of the method and analysis used for the adhesiveness test, which might have led to high fluctuation and scatter in the recorded values. However, the high precision of the BE testing with respect to hardness, toughness and stickiness might explain the high correlation with sensory properties found by Jindal and Limpisut [27] and Cheevitsopon et al. [15], and with the incubation times described in this study. The characteristics of brown rice that has germinated are influenced by the incubation time. Cho et al. [28] investigated the effect of germination on the physicochemical and textural properties of brown rice (BR) in different rice varieties (Samkwang, Misomi, Chindeul, and Hyeonpum). The results showed that hardness and toughness were decreased by germination, whereas stickiness and adhesiveness increased significantly. These results revealed that germination leads to improvements in the cooking and eating properties of BR.

### 3.2. Effects of Different Soaking and Incubation Durations on Cooked GBR Texture in the Production of Khao Dawk Mali 105 (KDML 105) GBR 

With a soaking time of 24 h, there were no correlations between incubation times and the texture properties of hardness, toughness, stickiness and adhesiveness tested by BE; the coefficient of determination (R^2^) values for these properties were 0.0357, 0.0087, 0.0064 and 0.1452, respectively. This indicated that the soaking time was too short to result in any linear variation in texture, even with a wide range of incubation times from 0 to 36 h. However, the texture of cooked GBR was softer, a result in line with the findings of Chao et al. [29], who found that the hardness of grains soaked for 12 h and germinated for 30 h was on average 24 N (39%) lower than that of brown rice. Paddy soaking in water at ambient temperature (20–30°C) requires 36 to 48 hours for a 30% moisture content level to be obtained. A soaking time less than this may not affect texture properties. 

With a soaking time of 48 h, the R^2^ relationships between incubation times and hardness, toughness, stickiness and adhesiveness were 0.8182, 0.8054, 0.6396 and 0.0312, respectively (Figure 2). These characteristics are all affected by the ratio of the starch constituents amylose and amylopectin. Jiamjariyatam et al. [30] reported that a higher amylose content and longer incubation time resulted in a greater hardness in puffed products made with rice starch. This might also apply to the cooked rice KDML 105 used in the present study.

We found no relationship between adhesiveness and incubation time. However, hardness and toughness both decreased with incubation times, and stickiness increased with longer incubation periods. In line with our findings, Jindal and Limpisut [27] reported reliable empirical models developed for estimating sensory hardness and stickiness with R^2^ = 0.96, and an overall acceptability with R^2^ > 0.71 from the BE force (hardness by BE) and water-to-rice ratio as independent variables. Our findings were also confirmed by the work of Cheevitsopon et al. [15], who reported a correlation coefficient (R) of −0.856 for the hardness–softness of cooked rice using a sensory test and hardness by BE. 

Munarko et al. [31] reported that five varieties of Indonesia GBR experienced reductions in peak viscosity, trough viscosity, breakdown, setback, and final viscosity. The decrease in peak viscosity was attributed to the activity of endogenous hydrolytic enzymes such as amylase, which hydrolyzes starch to smaller molecules [32]. Both α-amylase and β-amylase increased as germination progressed, leading to a decrease in peak viscosity [31,33,34]. GBR peak viscosity might be related to the stickiness of cooked GBR which increases (becomes more viscous) as germination progresses, i.e., the incubation time increases (increased peak viscosity). Li et al. [35] reported that germination led to a decrease in amylose content, while the molecular weights of the germinated starches showed no significant changes; however, the relative crystallinity of grain starches decreased significantly during germination, and brown rice starches exhibited marginal increases in peak viscosities during germination. But Cho et al. [29] reported that the viscosity of germinated brown rice decreased and that germination percentages were linearly associated with reduced pasting characteristics (final, peak and setback viscosities); in addition, varieties with faster germination speed tended to have lower viscosities. These results suggest that the germination of certain varieties greatly reduced the final viscosity of the flour and the hardness of the cooked brown rice. Chao et al. [10] reported that early harvest and germination resulted in decreased pasting viscosities and cooked-grain hardness. A reduction in setback value indicates that GBR is more stable against retrogradation [31]. This explains why a decreased hardness in cooked GBR (i.e., a softer product) might be due to longer incubation duration causing retrogradation of the cooked GBR.

In the study of Oliveira et al. [36], the incubation process caused protein levels in GBR to decline due to hydrolysis of proteins to amino acids. Amylose was degraded and reduced in size, and starch granules had pitted surfaces due to the degradation of protein and starch. In another study, monounsaturated fatty acids (MUFAs) decreased while saturated fatty acids (SFAs) and polyunsaturated fatty acids (PUFAs) increased; however, total fat values did not change; they were transformed only in terms of their reactions [37]. Additionally, content levels of phytic acid, pyridoxine, niacin and thiamine contents have been found to decrease in brown rice after germination [38,39]. Dextrins and oligosaccharides are the primary breakdown products of both amylose and amylopectin when they are broken down by-amylase. The breakdown of starch also results in an increase in reducing sugars, a drop in total starch content, and a decrease in the pasting viscosity of germinated rice. The authors of [40] found that germination time (incubation time) was positively connected with the stickiness, sweetness and softness of cooked rice, but negatively correlated with hardness, cooking time, water uptake, and volume expansion [40]; these findings corresponded to the results of the present study.

According to Oliveira et al. [36], Rusydi Megat et al. [37], Jiamyangyuen and Ooraikul [40] and the study described in this paper, the reduction in protein, the degradation of amylose, and the decrease in MUFA, while SFAs and PUFAs increased, might be related to the decrease in hardness and toughness and the increase in stickiness. Jiamyangyuen and Ooraikul [40] found that a decrease in the pasting viscosity of germinated rice due to the breakdown of starch was related to the linear change in these texture properties.

To the best of our knowledge, no work has yet described the phenomenon of adhesiveness correlating with changes in the constituents of GBR during incubation. In the present study, adhesiveness fluctuated with different incubation times (R^2^ was very small) (Figure 2d). In future studies, researchers may wish to consider how the adhesiveness of cooked rice or cooked GBR affects the attachment of rice to production equipment during mixing or stirring processes, with consequent impacts upon the operational time and cost during production.

### 3.3. The Sensitivity of BE Test on Texture Properties of Cooked GBR Rice

Figure 3 shows the textural properties—hardness, toughness, stickiness and adhesiveness—of cooked GBR of 32 different commercial brands in Thailand using some different varieties and some same varieties. Ranges of values for means, standard deviations and coefficients of variation are shown in Table 5. The coefficient of variation shows the degree of variation of the properties and can show the variation relativity of different sets of data even when means differ dramatically. The BE method for measuring the hardness, toughness, stickiness and adhesiveness of cooked GBR produced different degrees of variation within the same sample set, indicating different levels of sensitivity to different properties. A high coefficient of variation, where relatively wide variations in the absolute value of the property can be detected, indicates high sensitivity when small resolutions can be detected, and vice versa. The sensitivity of the BE tests for stickiness, toughness and hardness all ranked higher, in that order, than the sensitivity of the method for adhesiveness, which ranked lowest. This implies that BE testing is sensitive for measurements of stickiness, toughness and hardness. It can be seen in Figure 3 that, in the production of GRB, KDML 105 was used either on its own or mixed with other varieties. A total of 13 brands of KMDL 105 were used out of the 32 brands, Thai jasmine rice; fragrant rice (KDML 105 is one variety of it) was used in 12 brands and a further 7 brands used other fragrant rice, mountain rice and color rice varieties. Therefore, the texture properties of these cooked GBR products did not differ significantly overall. Brands using Thai jasmine rice mixed with Red Hawm rice and Homnil (m27), Red Hawm rice (m28) and Red Cargo rice (m31) had the highest values for hardness and toughness. However, brands including Red Hawm rice (m17, m18, m19 and m20) were characterized by a low hardness and low toughness, perhaps as a result of their different production methods. The softest and most easily deformed (least tough) cooked GRB was from KDML 105 (m2), the adhesiveness of which was also the lowest; it was also classified among the least-sticky group.

Table 6 shows the correlation between the texture parameters of commercial GBR in Thailand; these indicate the highest correlation coefficient was between hardness and toughness; other parameters correlated reasonably well with adhesiveness, but stickiness was not well correlated with other parameters. These results confirm that there is no relationship between the coefficient of variation (sensitivity) and the correlation between texture properties.

## 4. Conclusions

In this study, we carried out an evaluation of the precision and sensitivity of the back extrusion (BE) test for measuring the texture of cooked germinated brown rice in a production process. Our first objective was to evaluate the effects of different soaking and incubation times in the production of Khao Dawk Mali 105 (KDML 105) GBR on the cooked GBR texture, as measured by BE testing. We found that a 24 h soaking time was too short to produce any linear variation in texture, even when incubation times ranged from 0 to 36 h. However, a soaking time of 48 h resulted in the hardness and toughness both decreasing with incubation times, while stickiness increased with incubation times. There was no relationship between adhesiveness and incubation time. Our second objective was to determine the precision and sensitivity of BE testing of the textural properties of cooked GBR rice. We found that BE testing gave highly precise measurements of hardness, toughness and stickiness, which were both repeatable and reproductible, and highly sensitive measurements of stickiness, toughness and hardness, which were confirmed by the coefficients of variation in the textural properties measured using the BE test. These findings might explain why the texture properties measured using BE correlated well with the incubation times during soaking in the GBR production process. This idea is supported by the good correlation with the texture properties obtained using sensory testing, and by the change in constituents in GBR during incubation time reported by other researchers and referred to in the Results and Discussion section of this paper. However, the correlation coefficient among the texture properties by BE was not related to the precision or sensitivity of the test. In this study, we proposed an original protocol for the determination of the precision and sensitivity of food-texture measurement; the results described above confirm the usability of this protocol. We invite the researchers in food texture studies to use our protocol developed to check the precision and sensitivity of texture measurement test rig using 10-15 samples pririor to real experiment to be conducted.

## Figures and Tables

**Figure 1 foods-12-03090-f001:**
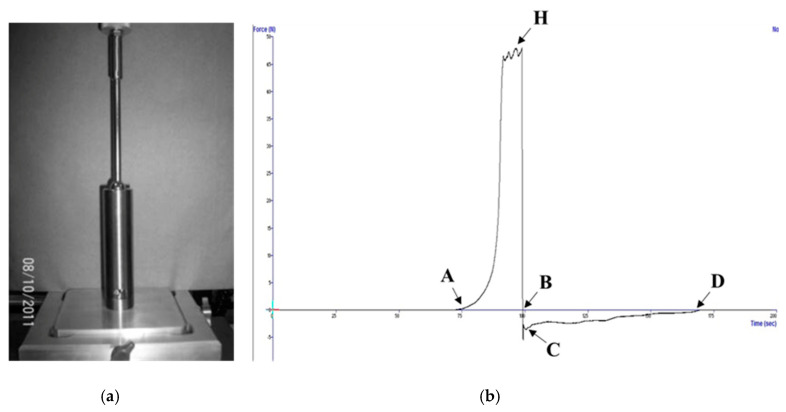
(**a**) Back extrusion (BE) test rig. (**b**) Force–time curve from cooked germinated brown rice BE test. (A) The point at which the stainless-steel ball probe first touched and began to compress the cooked germinated brown rice; (B) the point at which the stainless-steel ball began to withdraw from the compressed cooked rice, at a distance of 99 mm from the top opening; (C) the negative force (N) applied by the cooked rice to pull the probe, due to the adhesiveness of the former; (D) the point at which the probe wholly separated from the cooked germinated brown rice; and (H) the maximum compression force (N).

**Figure 2 foods-12-03090-f002:**
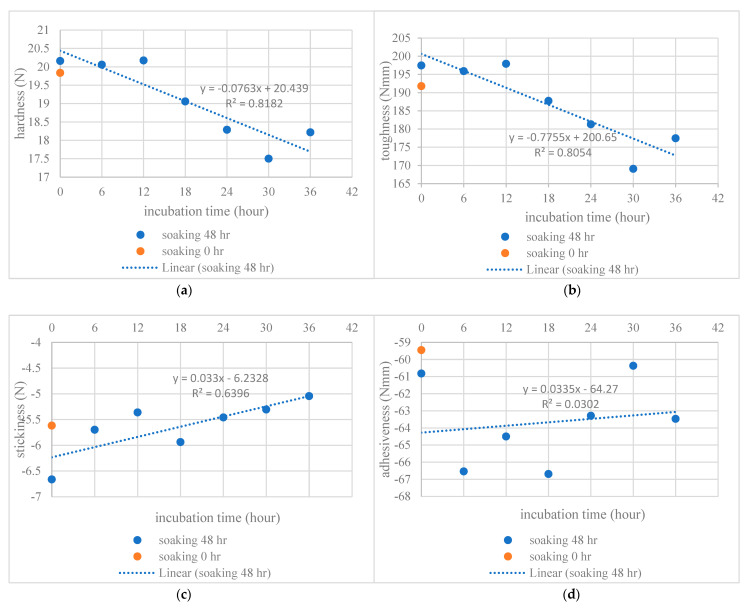
Effects of incubation time on the texture of cooked germinated brown rice: (**a**) hardness; (**b**) toughness; (**c**) stickiness; and (**d**) adhesiveness.

**Figure 3 foods-12-03090-f003:**
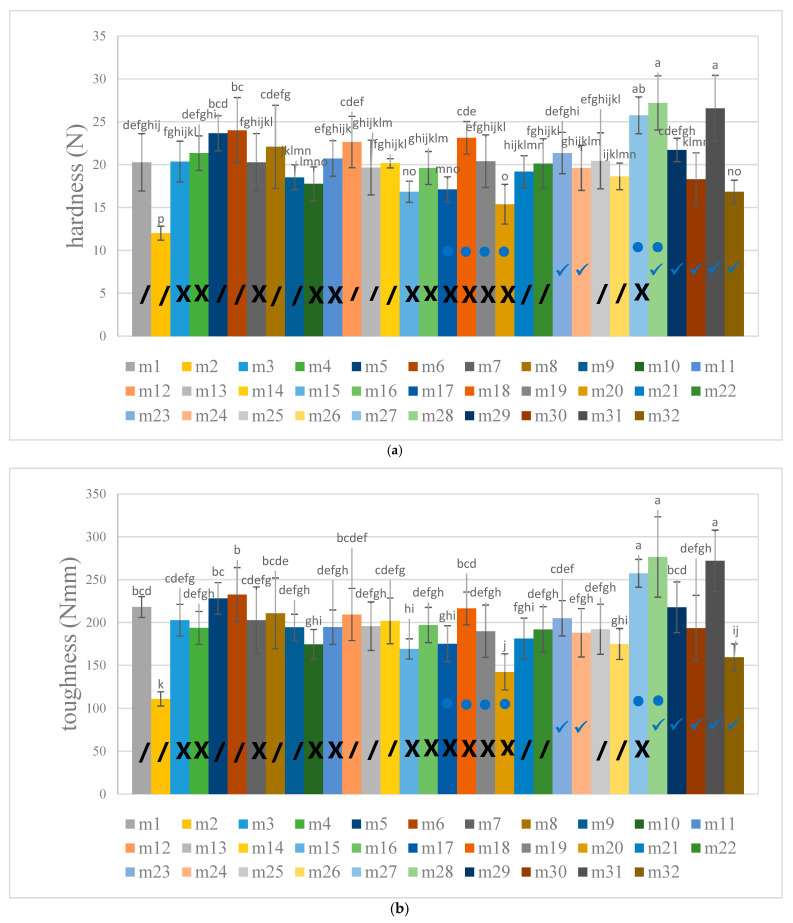
The texture properties of hardness (**a**), toughness (**b**) stickiness (**c**) and adhesiveness (**d**) of cooked germinated brown rice of different commercial brands in Thailand, with some varieties used alone or mixed with others. / indicates Khao Dawk Mali Rice (alone or mixed). 
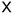
 indicates Thai Jasmine (alone or mixed). 
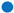
 indicates Red Hawm (alone or mixed). 
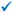
 indicates other varieties of fragrant, mountain and color rice. (m1) Khao Dawk Mali 105; (m2) Khao Dawk Mali 105; (m3) Thai jasmine rice; (m4) Thai jasmine rice; (m5) Khao Dawk Mali 105; (m6) Khao Dawk Mali 105; (m7) Thai jasmine rice; (m8) Khao Dawk Mali 105; (m9) Khao Dawk Mali 105; (m10) Thai jasmine rice; (m11) Thai jasmine rice; (m12) Khao Dawk Mali 105; (m13) Khao Dawk Mali 105; (m14) Khao Dawk Mali 105; (m15) Thai jasmine rice; (m16) Thai jasmine rice; (m17) Thai jasmine rice, Red Hawm Rice; (m18) Thai jasmine rice, Red Hawm Rice; (m19) Thai jasmine rice, Red Hawm Rice; (m20) Thai jasmine rice, Red Hawm Rice; (m21) Khao Dawk Mali 105, Red Hawm Rice, Hom Dang Sukhothai, Sinlek Rice, Homnil, Riceberry, Thai Pathumthani fragrant rice, Black sticky rice, RD6; (m22) Khao Dawk Mali 105, Homnil, Red Cargo rice; (m23) species not specified; (m24) Doi rice, Red Doi rice, Kum Doi Rice, Saren Rice, Ubon Ratchathani Hommali Rice; (m25) Khao Dawk Mali 105, Red Hawm Rice, RD6; (m26) Muser Purple Rice, Red Hawm Rice, RD6, Khao Dawk Mali 105; (m27) Thai jasmine rice, Red Hawm Rice, Homnil; (m28) Red Hawm Rice; (m29) Homnil; (m30) Homnil; (m31) Red Cargo rice; (m32) Muser Purple Rice.

**Table 1 foods-12-03090-t001:** Repeatability and reproducibility of reference tests for the hardness of GBR samples (N).

Repeatability SD				Reproducibility SD			
Sample Numbers	Duplicate.a	Duplicate.b	Diff. a-b	Sample Numbers	Duplicate.a	Duplicate.b	Diff. a-b
5, 17	21.16	21.90	−0.74	11, 18	19.49	20.89	−1.40
28, 33	19.83	18.54	1.29	28, 34	18.81	18.54	0.27
38, 57	12.75	12.11	0.64	44, 58	25.02	23.46	1.56
35, 69	20.57	22.17	−1.60	47, 70	21.30	19.68	1.62
		Mean	−0.11			Mean	0.51
		SD	1.31			SD	1.42

a and b is a replication number.

**Table 2 foods-12-03090-t002:** Repeatability and reproducibility of reference tests for toughness of GBR samples (Nmm).

Repeatability SD				Reproducibility SD			
Sample Numbers	Duplicate.a	Duplicate.b	Diff. a-b	Sample Number	Duplicate.a	Duplicate.b	Diff. a-b
5, 17	211.90	218.19	−6.28	11, 18	187.66	207.44	−19.78
28, 33	189.84	181.21	8.63	28, 34	181.45	181.21	0.25
38, 57	131.20	109.69	21.51	44, 58	246.03	233.45	12.58
35, 69	208.77	217.08	−8.31	47, 70	199.07	201.73	−2.66
		Mean	3.89			Mean	−2.40
		SD	13.97			SD	13.34

a and b is a replication number.

**Table 3 foods-12-03090-t003:** Repeatability and reproducibility of reference tests for stickiness of GBR samples (N).

Repeatability SD				Reproducibility SD			
Sample Numbers	Duplicate.a	Duplicate.b	Diff. a-b	Sample Number	Duplicate.a	Duplicate.b	Diff. a-b
5, 17	−5.35	−6.55	1.20	11, 18	−5.74	−4.65	−1.09
28, 33	−5.76	−5.91	0.15	28, 34	−6.64	−5.91	−0.73
38, 57	−4.70	−4.11	−0.58	44, 58	−6.56	−5.79	−0.77
35, 69	−4.50	−3.97	−0.54	47, 70	−4.61	−4.44	−0.17
		Mean	0.06			Mean	−0.69
		SD	0.83			SD	0.38

a and b is a replication number.

**Table 4 foods-12-03090-t004:** Repeatability and reproducibility of reference tests for adhesiveness of GBR samples (Nmm).

Repeatability SD				Reproducibility SD			
Sample Numbers	Duplicate.a	Duplicate.b	Diff. a-b	Sample Numbers	Duplicate.a	Duplicate.b	Diff. a-b
5, 17	−75.53	−69.92	−5.61	11, 18	−60.35	−74.47	14.11
28, 33	−63.87	−65.19	1.32	28, 34	−65.40	−65.19	−0.22
38, 57	−51.85	−50.69	−1.16	44, 58	−72.51	−71.20	−1.30
35, 69	−61.33	−59.73	−1.60	47, 70	−77.68	−60.51	−17.17
		Mean	−1.76			Mean	−1.15
		SD	2.87			SD	12.79

a and b is a replication number.

**Table 5 foods-12-03090-t005:** Statistics of texture parameters of commercial germinated brown rice in Thailand.

	Hardness (N)	Toughness (Nmm)	Stickiness (N)	Adhesiveness (Nmm)
Max	32.68	358.35	−1.86	−29.91
Min	10.83	100.90	−9.48	−97.85
Mean	20.39	198.95	−5.15	−66.45
SD	3.96	40.97	1.61	12.44
CV (%)	19.41	20.59	31.26	18.72

**Table 6 foods-12-03090-t006:** Correlations between texture parameters of commercial germinated brown rice in Thailand.

	Hardness	Toughness	Stickiness	Adhesiveness
Hardness	1			
Toughness	0.966	1		
Stickiness	0.206	0.191	1	
Adhesiveness	−0.693	−0.732	0.277	1

## Data Availability

Data are contained within the article.

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
