# Peer review of "Evaluation of Precision and Sensitivity of Back Extrusion Test for Measuring Textural Qualities of Cooked Germinated Brown Rice in Production Process"

_foods, 2023, doi:10.3390/foods12163090_

Round 1
Reviewer 1 Report
There are many points that need to improve
1- The abstract must contain numerical values
2-The objective of this work was to evaluate the effect of different soaking and aging duration( where in material and methods)
2-in figure 1 identify A, B,C, and D
3- the soaking methods do not mention in material and methods
4- statistical analysis must be added to material and methods
5-In Figure 2 d the R2 value is very small must be add comment
6- Figure 3 The significance must be added in each column and identify varieties that were used to possible used the best one( in methods and under figure)
Author Response
Thanks so much for your kind consideration for us to revise our manuscript. By comment you have given our manuscript is very much improved. Thanks so much.

Reviewer 2 Report
I enjoyed reading this submission. I was glad to see the authors took the time to do reproducibility and repeatability assays to determine where the variation originated and validate the instrumentation. I was left with a few questions after reading this.
Why was KDML 105 chosen for this study? This choice needs to be explained in more detail.
Grain aging effects usually occurs over months, not hours. I don’t understand how the aging effects would be observed so quickly. For Figure 2, is the aging time the amount of time the rice was soaked in water? If so, you should alter the labels for this figure. I don't understand how these data were collected and would like to see this explained further in the text.
Fixes needed in the ms:
1. What do the symbols (A-D,H) mean in Figure 1b? This should be explained in the legend.
2. Pg. 1, line 24: I think “sag”, should be sack or bag.
3. Figure 3 is unneeded. These data are summarized in the tables.
4. The data in Tables 1-4, don't quite fit on the page. I suggest reformatting so they do.
The text is easy to understand, but there are many instances where grammar or usage need to be corrected.
Author Response

(The authors gave the same response as above.)

Round 2
Reviewer 1 Report
Accept in present form